# Deep Reinforcement Learning for Engineering Design through Topology Optimization of Elementally Discretized Design Domains

**Nathan Brown [1], Anthony Garland [2], Georges Fadel [1], Gang Li [1]**

Clemson University [1], Sandia National Laboratory [2]

nkbrown@clemson.edu [1], agarlan@sandia.gov [2]

## Abstract

Machine learning (ML) can extract patterns in design-relevant data to detect trends or make predictions that may not be inherently visible to a human designer. However, most ML-based engineering design tools rely on supervised learning, which requires the prefabrication of design domain data that may be challenging to derive or inherently biased by a human designer. This work addresses these limitations by investigating the implementation of reinforcement learning (RL), a unique subset of ML that learns through accumulating past experiences in an interactive environment, into the engineering design problem of topology optimization. RL offers the design freedom and complexity of a human designer while maintaining the computational efficiency of more common machine learning paradigms. An RL environment was formatted to allow a deep RL agent to design 2D elementally discretized topologies based on a multi-objective reward function. After training, the agent was tested using progressive refinement on a variety of common load cases to validate the design capabilities and generalization of the agent. The results, which proved to be comparable to a traditional gradient-based topology optimization solver, show that a deep RL agent can learn generalized design strategies to satisfy multi-objective design tasks and, therefore, shows promise as a design tool for arbitrarily complex design problems across many design domains.

## Introduction

Machine learning (ML) can detect patterns within datasets and use those patterns to make predictions or perform sequential strategic decisions (Chi et al. 2021). Thanks to improvements in computational capabilities, new algorithms, and better transferring of data, ML techniques are more regularly being used to elevate the field of engineering design. ML models can act as integrated support for a human designer by extracting relationships and underlying trends within design-relevant data that the human designer may not intuitively discover. ML-based engineering design tools have led to improvements in additive manufacturing efficiency (Jiang et al. 2020), material composition design

(Wen et al. 2019), component design and optimization (Romeo et al. 2020), and model building (Yang et al. 2018).

Incorporating ML into the engineering design process introduces design automation to aid a human designer. This automation leads to enhancements in design accuracy and reliability while improving operational efficiency. Alternative design automation approaches can come in the form of objective-based optimization problems. These approaches rely on a design task being defined using an objective function that must be optimized given bounded parameters. Unfortunately, many design tasks cannot be defined using a function or may be subjected to severe suboptimal results given the bounded parameters.

ML researchers have successfully implemented ML into various engineering design domains (Jiang et al. 2020, Wen et al. 2019, Romeo et al. 2020, Yang et al. 2018, Liu et al. 2017). However, this paper will specifically focus on the design task of sequential topology design and optimization. Topology optimization (TO) attempts to optimally place material within a domain to minimize an objective while satisfying constraints based on weight and compliance (Sigmund and Maute 2013).

While ML has been used to improve the results and efficiencies within several engineering design domains, including TO (Wang et al. 2021), most ML-based design tools rely on supervised learning, a subset of ML that uses prefabricated data to map relationships between sets of inputs and outputs. These prefabricated datasets can be challenging to derive for particular design problems or inherently biased by the designer (Jordan and Mitchell 2015). Therefore, this research applies deep reinforcement learning (RL), a unique subset of ML that trains an agent by accumulating past experiences within an interactive environment, to design objective-based topologies. An RL agent uses past experiences to learn a set of actions to best achieve some objective, similar to how a human designer can use past experiences and

knowledge to perform a set of actions to design an object that best satisfies a design objective.

RL has been used to aid in engineering design and optimization, including microfluidic device design (Lee et al. 2019), microchip floor planning (Mirhoseini et al. 2021), 3D shape modeling (Lin et a.l 2020), and metasurface design (Sajedian, Lee, and Rho 2019). However, to the best of the authors' knowledge, there has only been one other successful attempt to use an RL agent to sequentially design optimal topologies. Hayashi and Ohsaki (Hayashi and Ohsaki 2020) successfully trained an RL agent to design optimal binary truss structures given TO objectives under various load cases. The results from this work proved the generalization capabilities of an RL agent when applied to truss-based TO. However, viewing the design domain as a truss structure severely limits the design complexity that can be achieved compared to an elementally discretized design domain.

This paper attempts to bridge this gap by proposing an RL-based topology designer built through sequential interactions with an elementally discretized topology. To the best of the authors' knowledge, this work is the first successful attempt to introduce an RL-based topology designer in such a design domain. This paper also takes a step towards validating that an RL agent can learn generic design strategies through interactions with a rich environment formatted to represent a particular engineering design problem.

## Topology Optimization as an RL Problem

The elementally discretized TO problem must be represented as a sequential RL task, specifically a Markov Decision Process (MDP). An MDP is a discrete-time stochastic process where state transitions and reward function solely depend on the current state and chosen action and are independent of the previous states and actions. An MDP is built upon four elements, state-space (S), action space, A, transition probability function (P), and reward function (R) (Bellman 1997). If the TO problem can be expressed in terms of S, A, P, and R, then designing optimal topologies can be an MDP.

The elementally discretized environment is represented in Figure 1. The 2D environment is comprised of N-by-N discrete elements. These discrete elements will either be viewed as material (light grey) or voided (white). In addition, certain elements and their corresponding nodes must be treated as bounded (black) or loaded (dark grey) to represent unique load cases.

### State Space

The state-space, S, of an RL environment represents all the combinations of observations that an agent can experience while interacting with the environment. Individual observa-

tions from the state-space are used to define the current representation of the environment. As the observation must be produced regularly during an RL task, the computational burden of generating the observation should be minimal.

Each observation is built as an NxNx3 3D array, with a mix of stress-based and boundary condition relevant information assigned at each element. The observations are variable in size depending on the desired topology size. The first NxN layer of the observation is the normalized inverse Von Mises Stress of each element. The Von Mises stress is commonly used within weight and compliance minimization TO problems to describe a topology's current stress state (Shimels 2017) and can be calculated using a simple 2D plane stress finite element analysis (FEA) solver. As the first layer uses the normalized *inverse* Von Mises stresses at each element, larger first layer values are associated with minimally stressed elements and zero values for any voided element.

The second layer of the observation is a Boolean representation of the elements viewed as bounded. If an element is bounded, the corresponding observation space will be assigned a value of 1, otherwise 0. Finally, the third layer is the loaded element equivalent of the second layer. An example of a simple 6x6x3 observation under a multi-loaded topology can be found in Figure A1 in Appendix A.

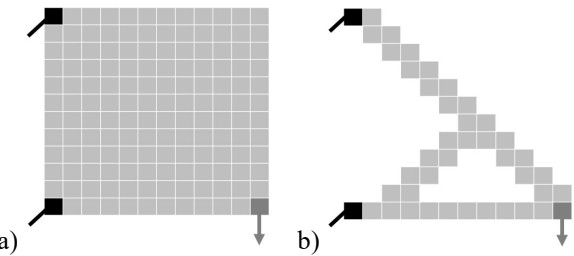

a)                                b)

Figure 1. 2D Elementally Discretized 12x12 Cantilever Beam a) Starting Topology b) Optimal Topology

### Action Space

The action space entails all the possible actions the agent can take given a current observation. Thus, actions describe how an agent can interact with an environment. In the TO environment, an action corresponds to voiding a single element, leading to a new topology and its corresponding observation. Within an NxN topology environment, the action space size is $N^2$, with each action corresponding to the voiding of a single element.

Designing topologies (Figure 2) is completed by sequentially selecting elements for removal from a starting solid block topology until an illegal action is taken or a termination criterion is met. The illegal actions, shown in Figure A2 of Appendix A, include trying to remove a bounded, loaded, or previously voided element or trying to remove an element that would lead to a non-singular body.

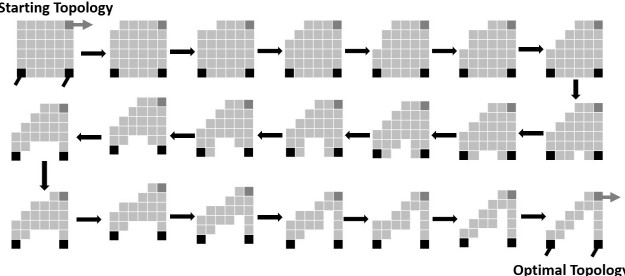

**Starting Topology**

**Optimal Topology**

Figure 2. Element Removal Sequence of 6x6 Topology

## Reward Function

The reward function controls the positive reward or negative penalty assigned to an agent after taking an action. The reward function should be formatted to ensure the environmental objective is being maximally satisfied. To satisfy the environmental objective of mimicking a TO solver, the authors built the multi-objective reward function to reward the agent for achieving topologies with minimal weight and compliance. In the 2D topology representation, the weight is represented as the current volume fraction (VF), and compliance is represented as the strain energy (SE). The VF will automatically decrease after any element is removed, and, therefore, the agent should be assigned more reward if it removes an element that leads to a minimal increase in SE compared to the starting solid block topology. The reward function is defined in Equation 1.

$$Reward = 5 * \left(\frac{SE_{Initial}}{SE_{Current}}\right)^2 + 5 * (1 - VF)^2 \quad (1)$$

where $SE_{Initial}$ is the SE of the starting solid block topology, $SE_{Current}$ is the SE of a resulting topology after the agent selects an element to remove, VF is the current volume fraction of the solid elements. A graphical representation of this multi-objective reward function is found in Figure A3 in Appendix A. Equation 1 was only used if the agent takes an action deemed "legal." If one of the illegal actions in Figure A2 is taken, the agent is penalized -1.

## Transition Probability Function

The transition probability function, also called the value function, selects the agent's action given a current observation and the predicted value for each action. The value function is built on a deep neural network using four convolution layers, with 16, 8, 4, and 1 3x3 filters, respectively. Convolutional filters are not size-dependent and, therefore, can adjust to multiple topology sizes without retraining. The value function takes the environmental observation as an input and outputs the predicted value of each action in the action space. The TO environment's action and state spaces are too large for the value function to know the exact value of taking each action given each observation, and therefore, the deep neural network is necessary to predict the state-action values

(Popova, Isayev, and Tropsha 2018). The deep neural network architecture is shown in Figure A4 in Appendix A.

The value of each state-action pair is calculated using double Q-learning, an extension of generic Q-learning that helps combat overestimation and avoid sub-optimal policy convergence (Sutton and Barto 2015).

## Training the Agent

The previous section defined the TO problem within an RL environment, and therefore, the RL agent should be able to satisfy our design objectives. In order to achieve these objectives, the agent, specifically the deep neural network that defines the agent's transition probability function, must be trained through episodic interactions with the TO environment. An episode in this environment is defined as a sequence of element removal actions starting at the initial solid block topology and continuing until one of the illegal moves is made or a termination criterion is met.

All training was completed on a 6x6 topology to limit the computational cost of running the FEAs needed to produce the agent's observations at each step. The training sequence assigns the RL agent randomly generated load cases to ensure thorough state-space exploration. The load cases were introduced by randomly assigning two elements on the exterior edge of the topology to act as bounded elements and a single random element to serve as the loaded element. The loaded element was randomly assigned a horizontal or vertical and compressive or tensile load. The training was run for 5000 episodes, taking approximately 1.5 hours using a PC with Intel® Core ™ i7-10510U CPU @ 1.80 GHz and 16 GB of RAM. The successful training results are included in Figure B1 in Appendix B.

## Testing the Agent

The trained agent could now be tested to validate its accuracy and generalization capabilities. As previously stated, the agent was only trained to design on 6x6 topologies to limit the computational burden of training. Attempting to design optimal topologies at the coarse 6x6 level will not yield practical results. Therefore, testing was completed on finer mesh size. The RL agent can interact with other topology sizes because its value function is built solely on convolutional filters, which are not size-dependent. Progressive refinement was implemented to increase mesh complexity while maintaining computational efficiency.

Progressive refinement allows an increase in topology detail over sequential. (Kim and Weck 2005). As a result, the equivalent topology shape progresses from 6x6 to 24x24 using an intermediate 12x12 size. This progressive refinement transition improves the efficiency compared to starting at

the 24x24 size because the agent can start removing elements in the coarse 6x6 topology which corresponds to removing 4 and 16 elements in the 12x12 and 24x24 topologies, respectively. The agent interacts with the simpler 6x6 topology until an intermediate volume fraction (VF) is met, causing the agent to transition to the equivalent 12x12 shape. Additional elements at the 12x12 size, corresponding to 4 elements at the 24x24 size, are removed to form low-level details. Finally, the equivalent 24x24 topology is formed from which the agent can remove elements to produce the final detailed optimal topology. This process can be seen without the individual element removal actions in Figure 3.

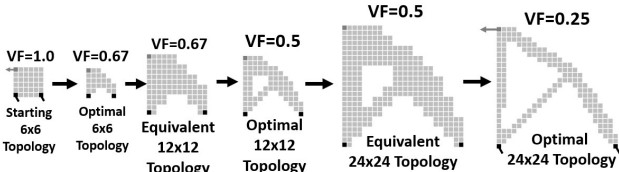

Figure 3. Sequential Elemental Removal with Variable Volume Fraction (VF) using Progressive Refinement

Several test cases have been presented to determine the accuracy and generalization capabilities of the presented method. Each test case introduces a unique combination of loaded and bounded elements and a user-specified final VF. The bounded and loaded elements were assigned based on their location within the 24x24 topology. During the progressive refinement steps, the equivalent elemental regions in the 6x6 and 12x12 topologies were treated as the loaded or bounded elements. The results in Table 1 show the final proposed 24x24 topology and intermediate 6x6 and 12x12 topologies for each test case. A more in-depth test case result table, Table C1, is available in Appendix C. Table C1 compares proposed topologies from the RL agent to those from a more traditional gradient-based TO solver, along with the final SE and VF of the topologies.

Tables 1 and C1 show that the RL agent has learned a generalized design strategy by satisfying the TO-based multi-objective reward function. Each proposed topology represents a feasible, objective-satisfying design solution.

The test cases validate that the RL agent has been able to generalize to the TO problem and did not memorize actions during training. The first indication of generalization is that the agent only interacted with 6x6 topologies during training, but during testing, the agent strategically removed elements on 6x6, 12x12, and 24x24 topologies without retraining. Thus, the agent used the same design strategy it learned during the 6x6 training and applied it to the larger environments. The second indication of generalization is that the last two test cases used multiple elements as loaded elements, whereas only a single element was randomly selected

during training. Therefore, the agent could not have experienced these load cases during training and must have generalized to unpredicted circumstances. These two experiences indicate that the agent's design strategy is general enough to account for unique, unseen load cases.

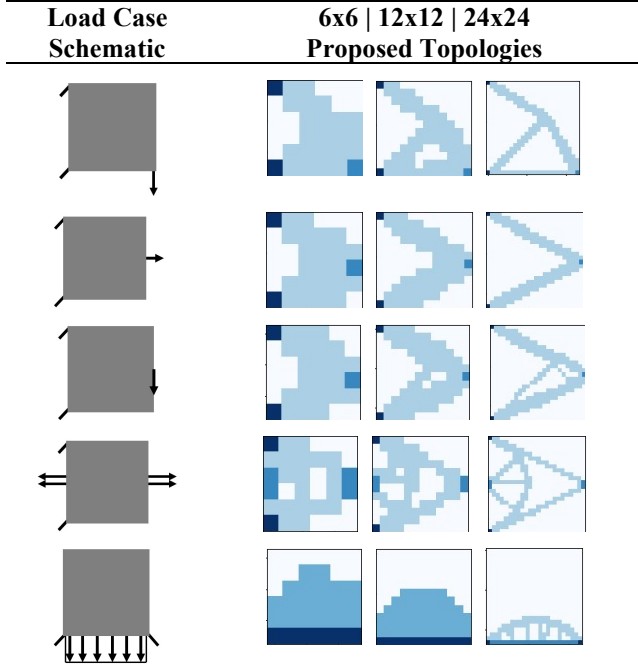

| Load Case Schematic | 6x6 \| 12x12 \| 24x24 Proposed Topologies | | |
|---|---|---|---|

Table 1. Test Case Results

## Conclusion

The presented work incorporated deep reinforcement learning into the optimization of 2D elementally discretized topologies. The reinforcement learning environment allowed an agent to sequentially remove elements from a topology and be rewarded for actions best satisfied the objective of a compliance minimization topology optimization problem. The agent, built on a deep neural network comprised of only convolutional filters, was trained to design optimal topologies at a 6x6 topology size with randomly selected loaded and bounded elements. During testing, a two-step progressive refinement approach was used to improve the detail representation of the topology from 6x6 to 12x12 to 24x24 without retraining the agent. The agent was tested on a diverse series of load cases. The results showed that after training, the reinforcement learning agent adopted a generalized design strategy that could design objective-based topologies. To the best of the authors' knowledge, this paper is the first successful attempt to design elementally discretized optimal topologies using reinforcement learning. This work is a crucial step towards validating reinforcement learning as a valuable engineering design tool.

# Appendices

## Appendix A: Topology Optimization as a Reinforcement Learning Problem Additional Figures and Comments

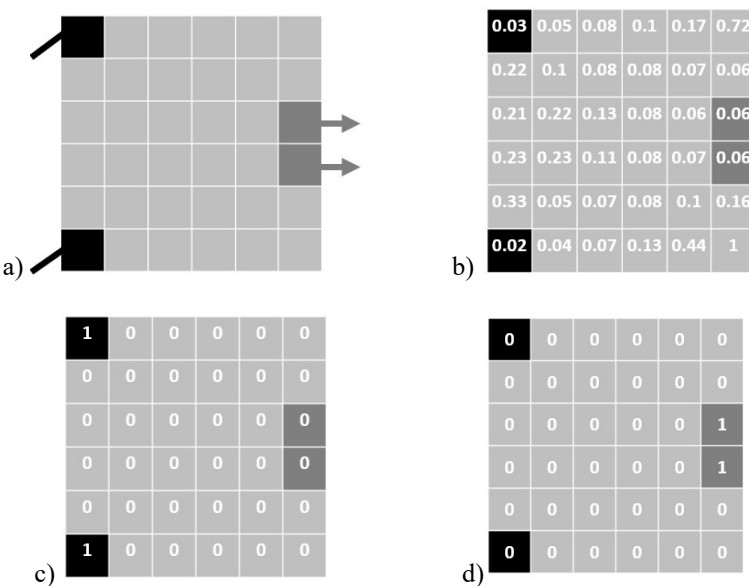

Figure A1. Observation of a Starting 6x6 Topology a) Schematic b) First Layer: Elemental Normalized InversVon Mises Stress c) Second Layer: Boolean Representation of Bounded Elements d) Third  Layer: Boolean Representation of Loaded Elements

The illegal actions, illustrated in Figure A2, include trying to remove a bounded, loaded, or previously voided element or trying to remove an element that would lead to a non-singular body. A non-singular body arises when the topology is held together by a hinge point. This single-node connection at a hinge point does not represent a feasible part of the design solution as it artificially inflates the stiffness of the topology while not representing a physical material connection. When an illegal action is made, the sequential design episode is terminated.

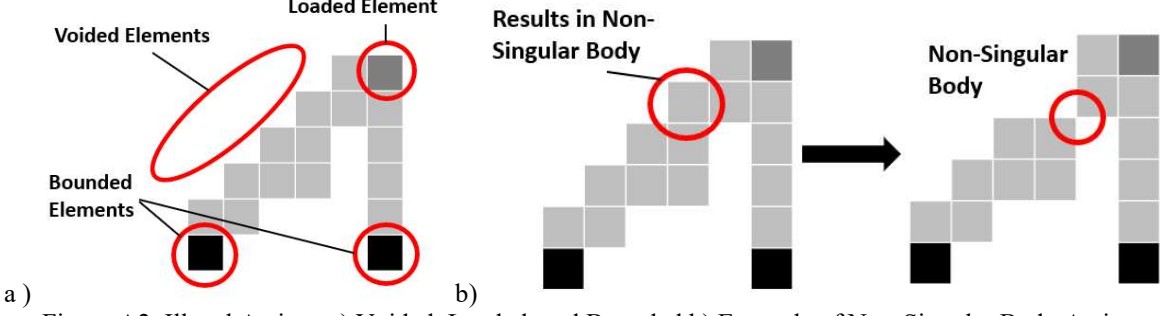

Figure A2. Illegal Actions a) Voided, Loaded, and Bounded b) Example of Non-Singular Body Action

The quadratic representation of the reward surface in Figure A3 was selected through experimentation to help magnify the optimal rewards during the later stages of the design process when the removal of various elements could lead to a similar strain energy (SE) increase.  These additions helped distinguish the optimal action from a group of similar yet slightly inferior actions. The authors of this paper do not claim this approach represents an optimal function for RL-based TO, but the results from this paper demonstrate the successful implementation of this multi-objective reward function.

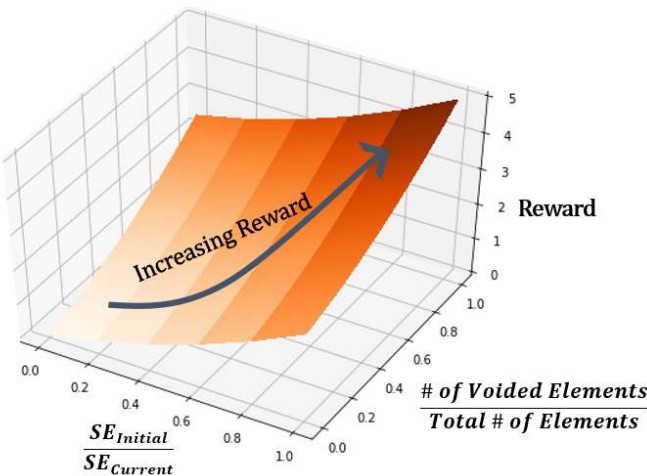

Figure A3. Graphical Representation of Multi-Objective Reward Function

The value function's deep neural network was built upon four convolutional neural networks (CNN) layers, each with a varying number of 3x3 convolutional filters. The purpose of a CNN is to extract relevant features from an input matrix and use those features to make decisions about the data. Feature extraction takes place by training the filters to recognize data relationships within an input. The initial layer filters capture generic, low-level features, whereas the later layer filters distinguish high-level detailed features. The CNN architecture, represented in Figure A4, shows the NxNx3 observation is used as the input and passed through four convolutional layers comprised of 16, 8, 4, and 1 convolutional filters, respectively, to produce an output of the predicted value of taking each action. The action associated with the highest value is selected, and therefore that element is removed, leading to a new observation. Convolutional filters are not size-dependent and can extract features from any input size and adjust to multiple topology sizes without retraining.

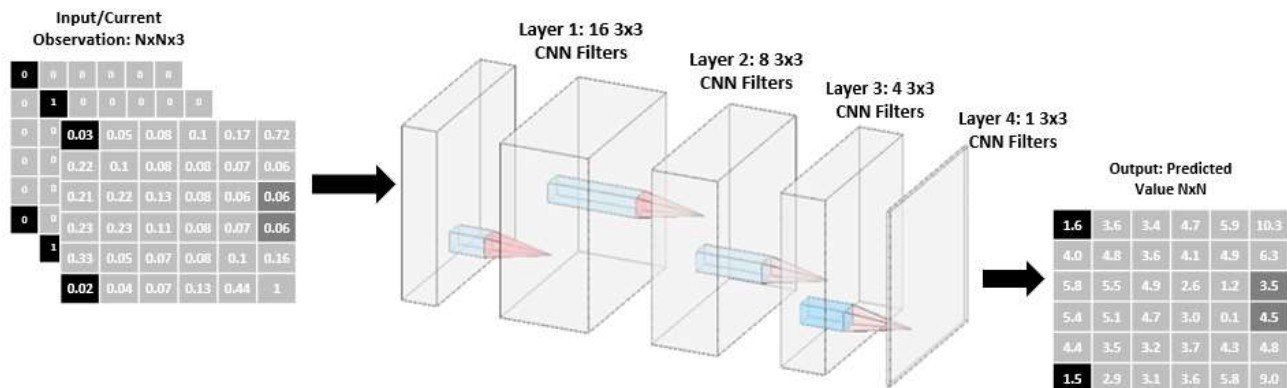

Figure A4. Deep Convolutional Neural Network Architecture

## Appendix B: Training the Agent Additional Figures and Comments

Figure B1 shows that the average training reward increases as training progresses. The agent was trained using the Epsilon-Greedy method. This method prompts the agent to start training by taking random actions to ensure the proper exploration of the observation space. As training continues, the agent takes fewer random actions and exploits the known value prediction from previous observations, leading to better actions and higher episodic rewards. The reward starts to generally converge towards a maximum as the agent takes nearly all exploitative actions and rarely takes explorative actions. The average reward in Figure B1 does not converge to a single value because the maximum reward the agent could accumulate during an episode changes depending on the load case and the corresponding number of elements that could be removed while not breaking the

non-singular body specification. The increase in reward in Figure B1 indicates the agent has learned to make more strategic actions that lead to design topologies that better satisfy the design objectives.

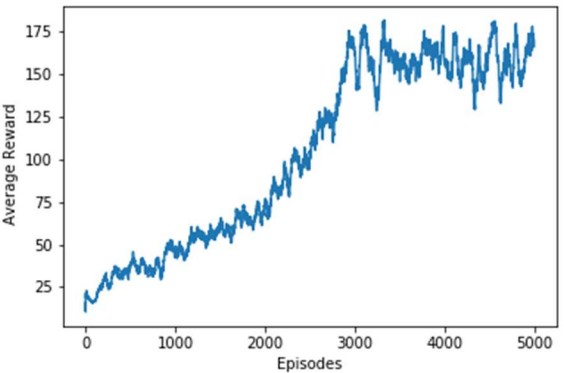

Figure B1. Average Reward During Training

## Appendix C: Test Case Detailed Result Table

| Load Case Schematic | 6x6 \|12x12 \| 24x24 Topologies | Final SE / Final VF | Gradient-Based Solver Equivalent | Gradient-Based Final SE / Final VF |
|---|---|---|---|---|
| A) | | $\dfrac{864}{0.25}$ | | $\dfrac{964}{0.25}$ |
| B) | | $\dfrac{193}{0.25}$ | | $\dfrac{211}{0.25}$ |
| C) | | $\dfrac{629}{0.31}$ | | $\dfrac{806}{0.25}$ |
| D) | | $\dfrac{1853}{0.25}$ | | $\dfrac{929}{0.25}$ |

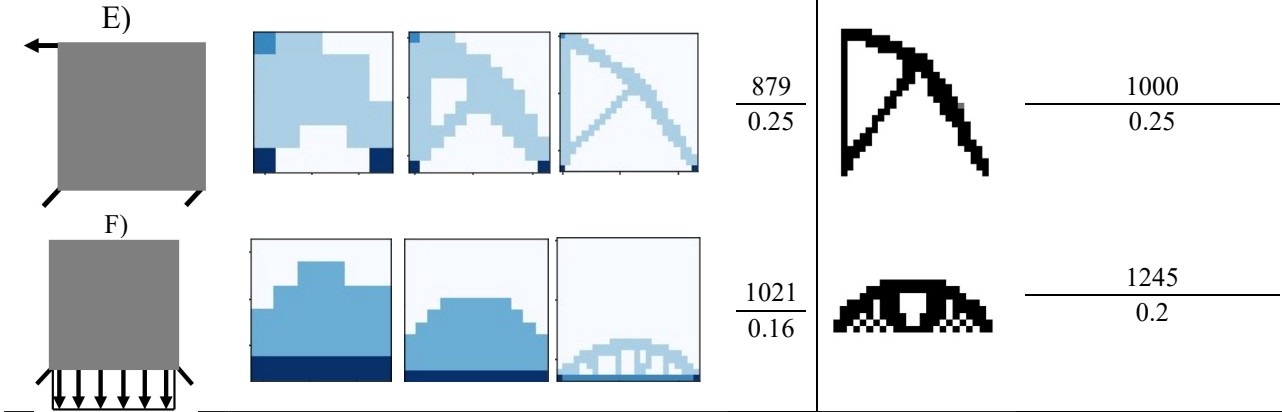

Table C1. Detailed Test Case Table showing the RL-based and traditional gradient-based (Sigmund 2001) proposed topologies. The final strain energy (SE) and volume fraction (VF) of both sets of proposed topologies are included. This comparison shows that the RL agent can design similar topologies as gradient-based solvers and can achieve reduced strain energies, better satisfying the design objective.

## Acknowledgments

The authors would like to acknowledge experimental facilities provided by the Center for Integrated Nanotechnologies (CINT). Sandia National Laboratories is a multimission laboratory managed and operated by National Technology & Engineering Solutions of Sandia, LLC, a wholly-owned subsidiary of Honeywell International Inc., for the U.S. Department of Energy's National Nuclear Security Administration under contract DE-NA0003525. This paper describes objective technical results and analysis. Any subjective views or opinions that might be expressed in the paper do not necessarily represent the views of the U.S. Department of Energy or the United States Government.

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
