# OpenReview forum: "Deep Reinforcement Learning for Engineering Design through Topology Optimization of Elementally Discretized Design Domains"
_AAAI.org/2022/Workshop/ADAM — AAAI 2022 Workshop ADAM_

### Official Review · Reviewer_uGAj · 2021-11-29
**Well-written paper with potential for extended results.  Minor clarifications will improve paper**

**Rating:** 8
**Confidence:** 4

**Review:**

This paper presents a reinforcement learning framework for solving the topology optimization problem. The authors demonstrate that by engineering the state space, action space, reward function, and the use of progressive refinement, their method can produce results that are competitive with conventional gradient-based methods while avoiding expensive FEA computations in the forward pass of the environment. They also show that their agent is able to generalize during test time to unseen loading conditions. The problem of topology optimization has been studied extensively in literature but the authors claim that this is the first work on using RL for topology optimization without any prior assumption on the design domain.  Overall, the paper is well-written, easy to follow along and the results are possibly significant if this method can be shown to be scalable to larger domains and also in 3D in the future. A few minor points that are confusing to be are:

What is the purpose of the coefficient value of 5 in the reward function if both the volume fraction and strain energy are multiplied?

It is not very clear how the observation and action space resolution changes are handled going from training to testing? For example, going from 6x6 to 12x12 will increase the action space dimension, and subsequently the output dimension of the policy from 36 to 144. Wouldn't that require changing the dimension of the final layer of the network?

---

### Official Review · Reviewer_x11B · 2021-12-02
**Well written paper on RL for TO**

**Rating:** 9
**Confidence:** 4

**Review:**

This paper presents a RL approach for solving TO problems in 2D. A sensible framing of the state space, action space is presented. By formulating the transition probability in terms of convolutional operators, the authors are able to smartly utilize a lower resolution FE solver to learn the transition probability. This along with progressive refinement results in a very competitive approach.

A couple of clarifications

- Can the authors show a result with different locations of the 2 bounded elements? While the authors claim that the bounded elements are randomly assigned, every result shown only has bounded elements on the two edges of the domain. In particular, I am interested in seeing what happens when the bounded elements are (a) diagonally placed, and (b) when they are placed close to each other

- How is the resolution jump handled? Please clarify